# Pharmacogenetics in Neuroblastoma: What Can Already Be Clinically Implemented and What Is Coming Next?

**DOI:** 10.3390/ijms22189815

**Published:** 2021-09-10

**Authors:** Gladys G. Olivera, Andrea Urtasun, Luis Sendra, Salvador F. Aliño, Yania Yáñez, Vanessa Segura, Pablo Gargallo, Pablo Berlanga, Victoria Castel, Adela Cañete, María José Herrero

**Affiliations:** 1Pharmacogenetics and Gene Therapy Platform, IIS La Fe, Hospital La Fe, Torre A-Lab 4.03, Av. Fernando Abril Martorell 106, 46026 Valencia, Spain; gladys_guadalupe@iislafe.es (G.G.O.); maria.jose.herrero@uv.es (M.J.H.); 2Department of Pharmacology, Faculty of Medicine, University of Valencia, Av. Blasco Ibáñez 15, 46010 Valencia, Spain; 3Pediatric Oncology Unit, Hospital Universitario y Politécnico La Fe, Av. Fernando Abril Martorell 106, 46026 Valencia, Spain; andrea.urtasun@sjd.es (A.U.); yanyez_yan@gva.es (Y.Y.); segura_van@gva.es (V.S.); pablogt28@gmail.com (P.G.); castel_vic@gva.es (V.C.); canyete_ade@gva.es (A.C.); 4Oncohematology Department, Hospital Sant Joan de Deu, Passeig Sant Joan de Déu 2, Esplugues de Llobregat, 08950 Barcelona, Spain; 5Department of Pediatric and Adolescent Oncology, Institute Gustave Roussy Center, Rue Edouard Vaillant 114, 94800 Villejuif, France; pablo.berlanga@gustaveroussy.fr

**Keywords:** SNP (single nucleotide polymorphism), chemotherapy, drug label, clinical implementation guidelines

## Abstract

Pharmacogenetics is one of the cornerstones of Personalized Precision Medicine that needs to be implemented in the routine of our patients’ clinical management in order to tailor their therapies as much as possible, with the aim of maximizing efficacy and minimizing toxicity. This is of great importance, especially in pediatric cancer and even more in complex malignancies such as neuroblastoma, where the rates of therapeutic success are still below those of many other types of tumors. The studies are mainly focused on germline genetic variants and in the present review, state of the art is presented: which are the variants that have a level of evidence high enough to be implemented in the clinic, and how to distinguish them from the ones that still need validation to confirm their utility. Further aspects as relevant characteristics regarding ontogeny and future directions in the research will also be discussed.

## 1. Introduction

Neuroblastoma (NB), the most common solid extracranial malignancy during childhood, has its origin in the adrenal medulla or paraspinal ganglia (sympathetic nervous system) during the period of development [1] and shows great phenotypic heterogeneity: some infants have spontaneous regression of the tumor while others present disease progression event after intensive multimodal treatment [2].

Although translational and clinical research has evolved considerably in recent years, in the cases requiring treatment, the prognosis of children with High-Risk Neuroblastoma (HR-NB) remains very poor. Despite the fact that it only represents around 8% of all pediatric cancers, it causes 15% of all deaths due to cancer in children. Barely 40% of the children diagnosed survive longer than 5 years [3,4]. Indeed, when metastatic relapse occurs, there is no curative treatment, and the overall survival rate after relapse is around 12 months. Thus, the lack of effective treatment continues to be a major concern for pediatric oncologists. Another relevant aspect to take into account is that >50% of survivor children face sequelae derived from chemotherapy toxicity along with their life, reinforcing the idea that chemotherapy needs to be optimized [5].

At this point, doctors need tools to apply the best available treatment to each patient in terms of maximizing efficacy and minimizing toxicity. Pharmacogenetics (PGx), one of the cornerstones of Precision Personalized Medicine, is the study of variants in the patient’s germline (constitutive) DNA related to the efficacy and safety of drugs. Tumor (somatic) genetic variants can also be considered, but they are out of the scope of this review. The most abundant genetic variants influencing PGx are the Single Nucleotide Polymorphisms (SNPs), accounting for approximately 90% of human genome variability [6,7]. However, key variants influencing PGx also include genomic insertions, deletions, and repeats, as well as genetic copy number variations (CNVs). 

The state of the art in PGx is divided into two categories: there is an already well-defined group of relevant associated drug-SNP pairs whose level of scientific-clinical evidence offers the highest quality, robust enough to be implemented in routine clinical practice. On the other side, drug-SNP pairs with less core evidence require further translational research to validate their usefulness in the real clinical setting.

In this review, the current situation of both categories is presented. The first part describes the genetic variants that can already be implemented in the clinical setting, and doctors can use them to adjust the NB chemotherapeutic treatment. After that, a view of the main variants that are currently under investigation to validate or discard their utility in NB treatment is presented. The last part of the review describes further considerations under a translational research perspective. The drugs included are those employed for (1) Induction treatments: cisplatin, vincristine, carboplatin, etoposide, cyclophosphamide, topotecan, doxorubicin; (2) Consolidation: busulfan, melphalan; and finally (3) Maintenance therapy and others: 13-cis-retinoic acid (isotretinoin), dinutuximab beta, interleukin 2, granulocyte and macrophage-colonies stimulating factor (GM-CSF) and radiotherapy.

## 2. Pharmacogenetic Variants for Clinical Implementation

Relevant institutions and scientific societies worldwide agree on the use of three main pillars as the references to identify those genetic variants with the highest level of evidence to be implemented in the clinic. These three cornerstones are the indications of drug regulatory agencies, PharmGKB (Pharmacogenomics Knowledge Base) [8], and relevant international consortia of experts developing clinical guidelines for PGx implementation.

### 2.1. Drug Regulatory Agencies

Regulatory agencies as the US Food and Drug Administration (FDA), the European Medicines Agency (EMA), the Pharmaceuticals and Medical Devices Agency from Japan (PMDA), and Health Canada Santé Canada (HCSC), amongst others [9,10,11,12], recommend in the Drug Labels a genetic test prior to the use of many drugs they approve. The Drug Label indicates if the test is required, recommendable, actionable, or simply informative [8]. At this point, it must be remarked that any requirement of a specific action in the drug label must be considered a legal requirement, with consequences in the case of disregard.

### 2.2. PharmGKB Clinical Annotations

PharmGKB, is a free access database created, curated, and managed by the University of Stanford and funded by US National Institutes of Health (NIH) and National Institute of General Medicine Science (NIGMS) [8]. It compiles most of the existing PGx information, from many different databases, including PubMed, under a Creative Commons license. It counts with a group of experts working on the dissemination of knowledge about the impact of human genetic variation on drug responses and on the translation of PGx into clinical practice. In fact, the website (www.pharmgkb.org, accessed on 25 June 2021) includes not only the information of its own curation from published data, the ‘Clinical Annotations’, but also that of the Drug Labels and the reports by Experts Consortia.

Clinical Annotations are the curated results obtained by PharmGKB’s experts after publications review for assigning different levels of evidence in the association of a genetic variant with efficacy, toxicity, metabolism/pharmacokinetics, and dosage of a drug. This ‘Level of Evidence’ ranks from 1 to 4, being 1 the one meeting the highest criteria. Level 2 is tagged as “moderate”, while 3 and 4 are “low” and “unsupported,” respectively. Very recently, these criteria have incorporated a new scoring system.

### 2.3. Clinical Implementation Guidelines

These are drug adjustment guidelines published by experts’ consortia to provide recommendations about what actions the prescriber should consider according to patient genotype. The main consortia are the Clinical Pharmacogenetics Implementation Consortium (CPIC) [13], the Royal Dutch Association for the Advancement of Pharmacy—Pharmacogenetics Working Group (DPWG) [14], but also other professional and scientific societies [15]. PharmGKB also compiles this information on the website and provides links to the complete articles under the section ‘Prescribing Info-Clinical Guideline Annotations’.

According to these three pillars, Table 1 shows the revised drugs that meet at least one criterion of the three pillars: PGx information in the drug label according to Drug Regulatory Agencies, PharmGKB Clinical Annotations with Level of Evidence 1 or 2, Clinical Implementation Guideline elaborated by an international expert Consortium. It must be underlined that, as previously stated, PharmGKB has just performed relevant modifications, beginning approximately on 25 March 2021. For caution, in this table, we have considered the immediate previous information regarding Clinical Annotations, but the current modifications have been marked.

Three drugs have genetic recommendations in Drug Labels according to different regulatory agencies, but these recommendations are not strictly PGx. They are mainly related to the use for which the drug is intended, its clinical indication, and the genes to be analyzed are not in the constitutive DNA of the patient but in the tumor. These drugs are Busulfan (FDA: *ABL1* and *BCR* genes), 13-cis-retinoic acid (not isotretinoin, but tretinoin has required testing for *PML* and *RARA* genes in the FDA Drug Label, the same in Health Canada Santé Canada Label, and also, but at an informative level, in the Japanese regulatory agency). The same situation applies to vincristine but, apart from Drug Label, it has a Clinical Annotation.

Dinutuximab contains information in the FDA label regarding *MYCN* in the clinical trials performed. 

In this sense, only the indications for *TPMT* regarding Cisplatin are really related to germline variants and a toxic event that could be prevented: ototoxicity.

Regarding Clinical Implementation Guidelines, we have this kind of source for two of the revised drugs: cisplatin and doxorubicin [16,17]. The recommendations of the CPNDS for the PGx-guided use of these drugs are summarized in Table 2.

## 3. Pharmacogenetic Variants under Investigation Regarding NB Therapy

Regarding research, there is still a lot of work to conduct. There are multitudes of SNPs whose influence on drug response was proposed but is not yet sufficiently tested, and, therefore, validation is needed. As shown in Table 1, only a few drugs from the group included in this review have one pillar (at least) supporting their implementation in the clinical setting. However, for the rest of the drugs, many of the genes coding the transporters, metabolizing enzymes, and targets of these drugs are already known, and research should be addressed to identify the implications of the SNPs in those genes on the safety and efficacy of those drugs. 

Table 3 shows a compilation of the most promising SNPs that are under research, according to the literature search. In most of the manuscripts, the studies have been performed with the combinations of drugs that are currently used in chemotherapy regimens. For clarity, we have searched for literature with results attributable to single drugs and not to combinations of them. Another criterion for the selection, in most cases, has been the known relationship between the gene and the drug in terms of transport, metabolism, and/or mechanism of action. 

Our knowledge about the genes involved in the mechanism of action, transport, and metabolism of the drugs is in many cases much more limited than expected. This happens not only with the newest drugs, but also with the classic chemotherapy that has been employed for decades. Therefore, focusing our attention on the SNPs contained in the genes responsible for the fate of the drug in our organism is not always easy. In addition, interactions exist, and these can happen immediately or with mid-long term effects, difficult to predict. All these together lead to “difficult to explain findings,” except for a much-reduced group of genes directly related to specific events. For example, SNPs in genes such as *MTHFR*, *TP53*, or *VDR* have been correlated with overall survival and event-free survival in PGx studies of NB patients, whereas these genes are not directly involved in the body routes of the chemotherapeutic drugs employed. In our group’s experience, rs1801133 in *MTHFR* (*p* = 0.02) and rs1544410 in *VDR* (*p* = 0.006) added an important predictive value for overall survival to the *MYCN* status, with a more accurate patients sub stratification than using *MYCN* alone [58,59,60].

If we check the bibliography, the most robust results should be reported from clinical trials, but these are obtained in predesigned and very much controlled situations different from the real clinical setting. For this reason, studies should be performed respecting the clinical reality, including concomitant treatments, especially considering that in pediatric oncology, many of the children participate in clinical trials evaluating their treatment and impeding them to participate in another specific for pharmacogenetics [61,62]. 

## 4. Further Consideration for a Perspective on NB PGx Translational Research

### 4.1. The Role of Ontogeny in NB Pharmacogenetics

The statement that children are not small adults is valid, particularly in pediatric clinical PGx. In order to offer children the optimal treatment, it is important not only to know the characteristics of the particular disease but also to integrate the changes of normal growth and development with their impact on the ontogeny of pharmacokinetic and pharmacogenomic factors. There are several age-related anatomic and physiological changes that have been found to influence drug ADME (Absorption, Distribution, Metabolism and Excretion) processes, such as differences in fat proportion in the body, gastric pH evolution, renal development, etc., but some of them are directly linked to the expression and role of relevant pharmacogenes. For instance, the activity of ABCB1 (P glycoprotein) and ABCG2, two of the most extensively studied ATP-binding cassette (ABC) transporters are decreased in the neonatal period [63,64]. Regarding metabolism, CYP3A7 shows the highest activity in the liver during embryonic, fetal, and newborn stages. After that, its activity declines, and other CYPs take the main roles. CYP3A4 appears during the first week after birth, reaching approximately 30–40% of adult activity by the first month and full adult activity by the 6th month of life. Its activity increases so much that it reaches 120% of the adults between 1 and 4 years of age, decreasing to normal adult levels after puberty [65,66,67]. In the case of CYP1A2, its expression is delayed until 3 months after birth. Regarding Phase II enzymes, as UGT1A family, it is remarkable that UGT1A1 starts to increase at birth and does not reach adult levels since 3–6 months later; and that UGT1A6 and UGT1A9 activity levels are smaller in people younger than 10 years old in comparison with adults [67]. 

### 4.2. What Else Do We Need to Take into Account? Research Integrating Epigenomics and Metabolomics 

The next goal is complementing PGx with two other relevant technologies that must undoubtedly be integrated: Epigenomics and Metabolomics. Explained in a very simple way, Epigenetics will deal with the study of the methylation status of the promoters of certain pharmacogenes as a mechanism of higher regulation that, above the nucleotide sequence of DNA, will determine whether a gene is really being expressed or not [68,69]. Metabolomics, also in a simplistic manner for this context, could characterize the plasma presence of metabolites corresponding to the drugs administered to the patients to check if the drugs have been effectively metabolized or not [70]. Complementing pharmacogenetics with the other two Omics techniques must help understand relevant questions: phenomena occurring in the patients that represent an upper layer of complexity for understanding the patients’ real response to drugs. For instance, metabolic pathways that are switched on when the main metabolic route is “off” or hyper/hypomethylation phenomena that regulate the expression of genes, masking potential effects of the nucleotide sequence. In order to understand the real characteristics of each patient, integration of all this information by means of very advanced Biostatistics, Systems Biology, Pharmacology, and Artificial Intelligence will be required. 

## 5. Conclusions

PGx knowledge needs to be implemented in the clinical routine of NB patients to support a more personalized approach regarding chemotherapy. The field is divided, with drug-genetic variants with a high level of scientific and clinical evidence and many more drug-SNP pairs needing further research in real patient contexts in order to validate their effects. The available literature regarding those associations is in many cases scarce and old, and the results have not been confirmed or updated. Thus, we need to put our efforts into this type of research. Meanwhile, those associations with high levels of evidence should be assessed in all our patients with the aim of providing the clinician with an additional tool to modify the treatment, if possible, and/or to be alert to increased risks of immediate or late toxicities. The current scenario provides data on relationships between SNP-drugs, but the reality is different most of the time due to the use of a combination of drugs. Thus, we need to validate the proposed “one-to-one” relationships in the real clinical context because interactions do exist, and the expected effects of concrete SNPs could not be the same in the context of polytherapy. 

## Figures and Tables

**Table 1 ijms-22-09815-t001:** Drugs included in NB treatment with clinically valuable PGx recommendations. The drugs included meet at least one of the clinical implementations three pillars-criteria: PGx included in Drug Label, PharmGKB Clinical Annotations Levels of Evidence 1 and 2, the existence of Clinical Implementation Guidelines published by relevant consortia. ALL: Acute Lymphoblastic Leukemia CPNDS: Canadian Pharmacogenomics Network for Drug safety, Cycloph: Cyclophosphamide, FDA: US Food and Drug Administration. Symbol − for guideline and drug label: lack of information.

Drug	Gene	SNP	Reference Genotype	Risk Genotype	ClinAnnot: Level	Recommendation for the Risk Genotype	Guideline/ Drug Label
Busulfan							−/FDA: Actionable *ABL1*, *BCR* (in ALL)
Carboplatin	*ERCC1*	rs11615	GG	AA, AG	2B: E,T	Moderate risk of inefficacy and toxicity	−/−
	*GSTP1*	rs1695	GG	AA, AG	2A: T	Moderate risk of toxicity	
	*MTHFR*	rs1801133	AA	AG, GG	2A: E	Moderate risk of inefficacy	
	*XRCC1*	rs25487	CC	CT, TT	2B: E	Moderate risk of inefficacy	
	*ERCC1*	rs3212986	AA	AC, CC	2B: T	Moderate risk of toxicity	
	*NQO1*	rs1800566	GG	AA, AG	2A: E	Moderate risk of inefficacy	
Cycloph.	*TP53*	rs1042522	CC	CG, GG	2B: E,T	Moderate risk of inefficacy and toxicity	−/−
	*SOD2*	rs4880	AA	AG, GG	2B: E	Moderate risk of inefficacy	
	*GSTP1*	rs1695	AA,AG	GG	2A: E,T	Moderate risk of inefficacy and toxicity	
Cisplatin	*TP53*	rs1042522	CC	CG, GG	2B: E,T	Moderate risk of inefficacy and toxicity	CPNDS: *TPMT*/ FDA: Informative *TPMT*
	*MTHFR*	rs1801133	AA	AG, GG	2A: E	Moderate risk of inefficacy	
	*GSTP1*	rs1695	AA	AG, GG	2B: T	Moderate risk of toxicity	
	*GSTP1*	rs1695	GG	AG, AA	2A: E	Moderate risk of inefficacy	
	*XPC*	rs2228001	TT	GT, GG	1B: T	High risk of toxicity	
	*XRCC1*	rs25487	CC	CT, TT	2B: E	Moderate risk of inefficacy	
	*ERCC1*	rs3212986	AA	AC, CC	2B: T	Moderate risk of toxicity	
	*ERCC1*	rs11615	GG	AA, AG	2B: E,T	Moderate risk of inefficacy and toxicity	
Doxorubicin	*NQO1*	rs1800566	GG	AA, AG	2A: E	Moderate risk of inefficacy	CPNDS: *RARG*, *SLC28A3*, *UGT1A6*/−
Etoposide	*DYNC2H1*	rs716274	AA	AG, GG	2B: T	Moderate risk of toxicity	−/−
Vincristine	*CEP72*	rs924607	CC,CT	TT	2B: T	Moderate risk of toxicity	−/FDA: Required *ABL1*, *BCR* (in ALL)

Note: the Clinical Annotations included in this table have been changed to Level 3 after new updates at www.pharmgkb.org from 25 March 2021.

**Table 2 ijms-22-09815-t002:** Summary of recommendations provided in the CPNDS guidelines for cisplatin and doxorubicin.

Gene	SNP	Reference Genotype	Risk Genotype	CPNDS Recommendations
CISPLATIN				
*TPMT*	rs1800462	CC	CG, GG	The consortium recommends testing these variants due to their relation with ototoxicity. ^+++^ Physicians are encouraged to consider the use of otoprotectors if the patient’s tumor type is one for which otoprotectors can be effective without adversely affecting antitumor activity. Alternative treatments may be prescribed when they have demonstrated equal efficacy, manageable and acceptable toxicity, less ototoxicity, and are considered options within the current standards of care. Increase monitoring in high-risk patients. Should be encouraged to receive more frequent follow-up audiometric hearing tests after treatment has ended.
	rs1800460	CC	CT, TT	
	rs1142345	TT	CT, CC	
DOXORUBICIN				
*RARG*	rs2229774	GG	AG, AA	The consortium recommends testing of these variants due to their relation with cardiotoxicity. ^+^ Increase frequency of monitoring, even with serial yearly echocardiographic monitoring and follow-up as recommended by COG guidelines; aggressive screening and management of cardiovascular risk factors, if the patient is considered at high risk. ^++^ Prescribe dexrazoxane. ^+++^ Use liposomal encapsulated anthracycline preparations; use of continuous inclusion or slower inclusion rates; use of less cardiotoxic types of anthracyclines; use of other cardioprotective agents; prescribe alternative chemotherapy regiments for certain tumor types where alternative regiments have been shown to be equally effective.
*SLC28A3*	rs7853758	AA, AG	GG	
*UGT1A6*	rs17863783	GG	GT, TT	

CPNDS: Canadian Pharmacogenomics Network for Drug Safety. Grading scheme used for clinical practice guidelines: ^+^ Level A corresponds to a high level of evidence (benefits clearly overcome the risks); ^++^ level B is a recommendation with lower scientific evidence level based on expert opinion; and ^+++^ level C is mainly based in expert’s opinion to be used in research context. SNP: single nucleotide polymorphisms, COG: Children’s Oncology Group.

**Table 3 ijms-22-09815-t003:** Candidate SNPs and genes to evaluate regarding the efficacy and toxicity of the drugs in Neuroblastoma (based in PharmGKB and the included references).

Drug.	Gene	SNP	Hypothetic Effect	References
Busulfan	*CTH*	rs1021737	Pediatric patients with the TT genotype (receiving hematopoietic stem cell transplantation) may have an increased risk for sinusoidal obstruction syndrome (SOS) when treated with cyclophosphamide and busulfan as compared to patients with the GG or GT genotypes.	Huezo-Diaz Curtis P., 2018 (Ref. [18])
	*CYP2C9*	rs1799853	Pediatric patients with the CT and TT genotypes (undergoing hematopoietic stem cell transplant) may have decreased metabolism of busulfan as compared to patients with the CC genotype.	Uppugunduri CR., 2014 (Ref. [19])
	*CYP2C19*	rs12248560	Pediatric patients with the CC genotype (undergoing transplantation) may have decreased metabolism of busulfan as compared to patients with the CT or TT genotypes.	
	*GSTA1*	rs3957357	Pediatric patients with the AG and GG genotypes (who are undergoing hematopoietic stem cell transplantation) may have decreased clearance of busulfan as compared to patients with the AA genotype.	Ten Brink MH., 2013 (Ref. [20])
	*GSTM1*	rs3754446	Patients with the AA and AC genotypes (and acute myeloid leukemia) may have decreased clearance of busulfan as compared to patients with the CC genotype.	Yee SW., 2013 (Ref. [21])
Carboplatin Cisplatin	*AKT1*	rs2494752	Patients with the GG and AA genotypes who are treated with carboplatin or cisplatin may have decreased risk of progression of the disease as compared to patients with the AG genotype.	Xu X., 2012 (Ref. [22])
		rs1130214	Patients with the CC genotype (and lung cancer) who are treated with carboplatin or cisplatin may have a higher risk of distant disease progression as compared to patients with the AC or AA genotype.	Xu JL., 2012; (Ref. [23])
	*EIF3A*	rs3740556	Patients with the GG genotype (and lung cancer) may have a poorer response when treated with platinum-based chemotherapy as compared to patients with the AA or AG genotype.	Xu X., 2013 (Ref. [24])
	*MTR*	rs1805087	Pediatric patients with the GG genotype and cancer may have an increased risk for drug toxicity and an increased response to treatment with cisplatin or carboplatin as compared to patients with the AA or AG genotypes.	Patiño A., 2009 (Ref. [25])
	*PIK3CA*	rs2699887	Patients with the CC genotype (and non-small-cell lung cancer) may have an increased risk for toxicity when treated with platinum-based chemotherapy as compared to patients with the TT genotype.	Pu X., 2011 (Ref. [26])
	*PTEN*	rs2299939	Patients with the AA genotype (and non-small-cell lung cancer) may have an increased risk for toxicity when treated with platinum-based chemotherapy as compared to patients with the AC or CC genotype.	
	*SLC31A1*	rs7851395	Patients with the AA genotype may have increased overall survival when treated with carboplatin or cisplatin (in people with Non-Small-Cell Lung Carcinoma) as compared to patients with genotypes AG or GG.	Xu X., 2012 (Ref. [22])
	Unknown (Intronic)	rs2498804	Patients with the CC genotype (and non-small-cell lung cancer) may have an increased risk of distant disease progression when treated with platinum-based chemotherapy as compared to patients with the AA or AC genotypes.	Pu X., 2011 (Ref. [26])
Cyclophosph.	*ABCB1*	rs1045642	Allele G is associated with an increased risk of death when treated with cyclophosphamide in combination with other drugs, (in patients with osteosarcoma) as compared to allele A.	Caronia D., 2011 (Ref. [27])
	*ABCC4*	rs9561778	Patients with the TT or GT genotypes (and breast cancer) who are treated with cyclophosphamide may have an increased risk of neutropenia/leukopenia and gastrointestinal toxicity, as compared to patients with the GG genotype.	Low SK., 2009 (Ref. [28])
	*CYP2B6*	rs7254579	Patients (with lupus) and the CC or CT genotypes may have decreased metabolism of cyclophosphamide, resulting in decreased concentrations of active cyclophosphamide metabolite as compared to patients with TT genotype.	Su W., 2016 (Ref. [29])
		rs4802101	Patients with the CC genotype may have decreased metabolism of cyclophosphamide, resulting in decreased concentrations of active cyclophosphamide metabolites and decreased risk of gastrointestinal toxicity, or leukopenia, as compared to patients with the CT or TT genotypes.	
		rs8192709	(Recipients of HLA-identical hematopoietic stem cell transplantation) with the TT or CT genotypes (and leukemia) may have an increased risk for hemorrhagic cystitis when treated with cyclophosphamide compared to patients with the CC genotype.	Rocha V., 2009 (Ref. [30])
		rs2279343	Patients with the GG or AG genotypes (who have received a hematopoietic stem cell transplant) and are treated with cyclophosphamide may have an increased risk for oral mucositis as compared to patients with the AA genotype.	
		rs3745274	(Leukemia patients who are) recipients (of HLA-identical hematopoietic stem cell transplantation) from donors with the GG genotype may have an increased risk of developing veno-occlusive disease of the liver when treated with cyclophosphamide as compared to donor cells with the GT or TT genotype.Patients with the GG or GT genotypes (and Breast Cancer) who are treated with cyclophosphamide and doxorubicin may be more likely to require a reduction in dose as compared to patients with the TT genotype.	Bray J., 2010 (Ref. [31])
	*CYP2C19*	rs4244285	Patients with the GG genotype (and Systemic Lupus Erythematosus) who are treated with cyclophosphamide may have increased metabolism of cyclophosphamide, leading to higher concentrations of the active metabolite and an increased risk of toxicity (ovarian, gastrointestinal, or hematological) as compared to patients with the AA and AG genotype.	Su W., 2016 (Ref. [29])
	*CYP3A4*	rs2740574	Premenopausal patients with the TT genotype (and breast cancer) who are treated with cyclophosphamide may have a shorter period before chemotherapy-induced ovarian failure compared to patients with the CC or CT genotype.	Su HI., 2010 (Ref. [32])
Dinutuximab			SNPs reducing or impairing the expression of *GD2* could impede Dinutuximab efficacy.	Chen RL., 2000 Greenwood KL., 2018 (Refs. [33,34])
Doxorubicin	*ABCB1*	rs1045642 rs2032582 rs1128503	Patients harboring the CC-GG-CC genotypes had significantly lower peak plasma concentrations of doxorubicinol compared to patients who had TT-TT-TT genotypes.	Lal S., 2008 (Ref. [35])
	*ABCC1*	rs45511401	Patients with the TT or GT genotypes (and non-Hodgkin lymphoma) who are treated with doxorubicin may have an increased risk for cardiotoxicity as compared to patients with the GG genotype.	Wojnowski L., 2005 (Ref. [36])
	*ABCC2*	rs8187710	Patients with the AA or AG genotypes (and non-Hodgkin lymphoma) who are treated with doxorubicin may have an increased risk of cardiotoxicity as compared to patients with the GG genotype.	
		rs17222723	Patients with the AA or AT genotypes (and non-Hodgkin lymphoma) who are treated with doxorubicin may have an increased risk of cardiotoxicity as compared to patients with the TT genotype.	
	*CBR1*	rs9024	Patients with the GG genotype may have increased clearance of doxorubicin and decreased exposure to doxorubicin compared to patients with the AG genotype.	Lal S., 2008 (Ref. [37])
	*CBR3*	rs8133052	Patients with the AA genotype (and breast cancer) who are treated with doxorubicin may have decreased metabolism of doxorubicin and may have greater tumor reduction, but may also have increased severity of neutropenia as compared to patients with the GG genotype.	Fan L., 2008 (Ref. [38])
	*CYBA*	rs4673	Cancer patients with the AA or AG genotypes who are treated with doxorubicin may have an increased risk for cardiotoxicity as compared to patients with the GG genotype.	Megías-Vericat JE., 2018 and Wojnowski L., 2005 (Refs. [36,39])
	*GSTA1*	rs3957357	Patients with the GG and AG genotypes (and soft tissue sarcoma) may have a shorter progression-free survival time when treated with doxorubicin as compared to patients with the AA genotype.	Gelderblom H., 2014 (Ref. [40])
	*GSTP1*	rs1695	Patients with osteosarcoma and the GG or AG genotypes may be at an increased risk of developing leukopenia when treated with doxorubicin as compared to patients with the AA genotype.	Windsor RE., 2012 (Ref. [41])
	*RAC2*	rs13058338	Cancer patients with the TT or AT genotypes who are treated with doxorubicin or idarubicin may have an increased risk for drug toxicity as compared to patients with the AA genotype.	Megías-Vericat JE., 2018 and Wojnowski L., 2005 (Refs. [36,39])
Etoposide			All these relevant pharmacogenes *ABCB1*, *ABCC3*, *CYP3A4*, *CYP3A5*, *GSTP1*, *UGT1A1*, are known to be relevant in etoposide pharmacokinetics, thus studies validating their main SNPs regarding the drug’s toxicity and efficacy are needed.	Relling M.V., 1994; Zelcer N., 2001; Huang RS.,2007 (Refs. [42,43,44])
GM-CSF			Human GM-CSF receptor beta chain gene could be a good candidate to investigate the role of SNPs that could interfere the activation of the receptor by the drug.	Shen Y., 1992 (Ref. [45])
Interleukin2			SNPs in the gene coding for IL2 receptor could be very informative to assess the response to this treatment.	Ladenstein R., 2018; YamaneB.H., 2009; Shusterman S., 2019 (Refs. [46,47,48])
Melphalan			*ABCB1* and *GSTP1* are relevant pharmacogenes that seem to be involved in melphalan pharmacokinetics. Exploring their main SNPs could be of interest.	Karkey M.A., 2005 and Hodges L.M., 2011 (Refs. [49,50])
Radiotherapy	*CDK1*	rs10711	Patients with the GG or GT genotypes may have increased risk of pneumonitis when treated with radiotherapy (lung cancer) as compared to patients with genotype TT.	Pu X., 2014 (Ref. [51])
	*PRKCE*	rs11125035	Patients with the AA genotype may have increased risk of esophagitis when treated with radiotherapy as compared to patients with genotype TT or AT.	Pu X., 2014 (Ref. [51])
	*TANC1*	rs10497203 rs264631 rs264651 rs6432512 rs264588 rs264663 rs7582141	Patients with the CC or AC/CG or GG/GG/CT or TT/AA or AC/CT or TT/GT or TT genotypes, respectively in the referred SNPs (left column) (and prostate cancer) who are treated with radiotherapy may have an increased risk of late stage toxicity as compared to patients with the other possible genotypes.	Fachal L., 2014 (Ref. [52])
Topotecan	*ABCG2*	rs4148157	Pediatric patients with the GG genotype (and brain tumors) may have decreased absorption and lower concentrations of topotecan compared to patients with the AA and AG genotypes.	Roberts J.K.,2016 (Ref. [53])
Vincristine	*ABCB1*	rs1045642	Pediatric patients with the AA or AG genotypes (and acute lymphoblastic leukemia) who are treated with vincristine may have a decreased likelihood of event-free survival as compared to patients with the GG genotype.	Ceppi F.,2014 (Ref. [54])
		rs4728709	Pediatric patients with the GG genotype (and acute lymphoblastic leukemia) who are treated with vincristine may have an increased risk of grade 1–2 neurotoxicity as compared to patients with the AA or AG genotypes.	Ceppi F.,2014 (Ref. [54])
			*CYP3A5* seems to be the main metabolizing enzyme for vincristine, thus analyzing its main SNPs could be of interest	Egbelakin A.,2011 (Ref. [55])
Isotretinoin	*LEP*	rs7799039	These SNPs seem to correlate with the lipid disorders caused by the drug.	Khabour O.F.,2018 (Ref. [56])
			Other SNPs in relevant genes related to the mechanism of action of the drug could shed light for decreasing adverse side effects and increasing efficacy: *RXRA*, *JAK2*, *CDC25C*	Lee J.J,2011 (Ref. [57])

## Data Availability

The data reported is contained in the referenced papers as well as in PharmGKB website (www.pharmgkb.org).

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
