# Peer review of "Pharmacogenetics in Neuroblastoma: What Can Already Be Clinically Implemented and What Is Coming Next?"

_ijms, 2021, doi:10.3390/ijms22189815_

Round 1

Reviewer 1 Report

Olivera and co-authors present a review about pharmacogenetics in neuroblastoma. This is a very interesting subject that could really help in patients treatment. The review is concise but complete at the same time, and it gives a comprehensive overview on what we already know about this matter and what we should examine more in depth.

I just strongly encourage authors to submit the text to the review by an expert in English language because I’ve found a great number of oversights, syntax errors, construction of phrases not very appropriate, etc.

And what does it mean [cita] in line 176, page 10? And (6) as superscript at the end of line 209, page 11?

Author Response

Comments and Suggestions for Authors

Olivera and co-authors present a review about pharmacogenetics in neuroblastoma. This is a very interesting subject that could really help in patients treatment. The review is concise but complete at the same time, and it gives a comprehensive overview on what we already know about this matter and what we should examine more in depth.

I just strongly encourage authors to submit the text to the review by an expert in English language because I’ve found a great number of oversights, syntax errors, construction of phrases not very appropriate, etc.

Thanks for the suggestion. The entiremanuscript has been extensively reviewed to correct language

And what does it mean [cita] in line 176, page 10? And (6) as superscript at the end of line 209, page 11?

We are sorry, the indicated references have been included following the journal style within the revised version of the manuscript.

Reviewer 2 Report

Gladys G. Olivera et al.: Pharmacogenetics in Neuroblastoma: what can already be clinically implemented and what is coming next?

The authors performed a literature review on collected data of genetic changes (mainly Single Nucleotide Pleomorphisms) and their influence on chemotherapeutic drugs’ effects on neuroblastoma tumours and on toxicity to host. They collected and tabulated numerous drugs and genes/SNPs (completing with the normal variants) and the relevant references which make the paper useful for pediatric oncologists or researchers dealing with pharmacology. They grouped the changes according to the level of evidence, and also indicated the promising candidate gene/SNP combos which are potential targets of future studies or examinations.

As a consultant dealing with rather the diagnosis of pediatric neoplasms including neuroblastoma, I can not add too much to the clinical message, but as a regular participant of pediatric MDTs (oncoteams) I know, that high-risk neuroblastoma patients usually receive combined chemotherapies. My question is how do clinicians utilize the single drug-SNP interactions in their everyday practice, where combinations dominate. How do drug interactions influence the whole question?

Looking at the text itself, I noticed some typing errors or grammatic problems, which are:

  1. page 5, line 133: repeated words, „to be analyzed” 2x
  2. page 6, line 159: misspelling (typing error), chemotherapy
  3. table 3, cancer in Spanish (cáncer) several times
  4. page 10, line 182: misspelling, classes
  5. page 11, line 196: misspelling, maturation
  6. page 11, line 217: word order, „the presence of metabolites in plasma”

Author Response

Comments and Suggestions for Authors

Gladys G. Olivera et al.: Pharmacogenetics in Neuroblastoma: what can already be clinically implemented and what is coming next?

The authors performed a literature review on collected data of genetic changes (mainly Single Nucleotide Polymorphisms) and their influence on chemotherapeutic drugs’ effects on neuroblastoma tumours and on toxicity to host. They collected and tabulated numerous drugs and genes/SNPs (completing with the normal variants) and the relevant references which make the paper useful for pediatric oncologists or researchers dealing with pharmacology. They grouped the changes according to the level of evidence, and also indicated the promising candidate gene/SNP combos which are potential targets of future studies or examinations.

As a consultant dealing with rather the diagnosis of pediatric neoplasms including neuroblastoma, I cannot add too much to the clinical message, but as a regular participant of pediatric MDTs (oncoteams) I know, that high-risk neuroblastoma patients usually receive combined chemotherapies. My question is how do clinicians utilize the single drug-SNP interactions in their everyday practice, where combinations dominate. How do drug interactions influence the whole question?

The reviewer is totally right and in fact that point is one of the current main challenges in the field.  We have tried to state this by adding the following paragraph in the Conclusions section, page 11, line 231:

"The current scenario provides data on relationships between SNP-drug, but the reality is most of the times the use of a combination of drugs, so we need to validate the pro-posed “one-to-one” relationships in the real clinical context, because interactions do exist, and the expected effects of concrete SNPs could not be the same in a context of polytherapy"

Looking at the text itself, I noticed some typing errors or grammatic problems, which are:

  1. page 5, line 133: repeated words, „to be analyzed” 2x
  2. page 6, line 159: misspelling (typing error), chemotherapy
  3. table 3, cancer in Spanish (cáncer) several times
  4. page 10, line 182: misspelling, classes
  5. page 11, line 196: misspelling, maturation
  6. page 11, line 217: word order, „the presence of metabolites in plasma”

Thanks for the comment. We have corrected them appropriately. Sorry